# Novel Insights on the Role of Nitric Oxide in the Ovary: A Review of the Literature

**DOI:** 10.3390/ijerph18030980

**Published:** 2021-01-22

**Authors:** Maria Cristina Budani, Gian Mario Tiboni

**Affiliations:** 1Department of Medicine and Aging Sciences, University “G. d’Annunzio” Chieti-Pescara, Via Dei Vestini 31, 66100 Chieti, Italy; maria.budani@unich.it; 2Department of Medical, Oral and Biotechnological Sciences, University “G. d’Annunzio” Chieti-Pescara, Via Dei Vestini 31, 66100 Chieti, Italy

**Keywords:** nitric oxide, folliculogenesis, steroidogenesis, meiotic maturation, in-vitro fertilization (IVF), embryo development

## Abstract

Nitric oxide (NO) is formed during the oxidation of L-arginine to L-citrulline by the action of multiple isoenzymes of NO synthase (NOS): neuronal NOS (nNOS), endotelial NOS (eNOS), and inducible NOS (iNOS). NO plays a relevant role in the vascular endothelium, in central and peripheral neurons, and in immunity and inflammatory systems. In addition, several authors showed a consistent contribution of NO to different aspects of the reproductive physiology. The aim of the present review is to analyse the published data on the role of NO within the ovary. It has been demonstrated that the multiple isoenzymes of NOS are expressed and localized in the ovary of different species. More to the point, a consistent role was ascribed to NO in the processes of steroidogenesis, folliculogenesis, and oocyte meiotic maturation in in vitro and in vivo studies using animal models. Unfortunately, there are few nitric oxide data for humans; there are preliminary data on the implication of nitric oxide for oocyte/embryo quality and in-vitro fertilization/embryo transfer (IVF/ET) parameters. NO plays a remarkable role in the ovary, but more investigation is needed, in particular in the context of human ovarian physiology.

## 1. Introduction

Nitric oxide (NO) is a short-lived gaseous molecule functioning as an intra- and intercellular biological messenger [1]. In mammals, NO generation is due to the action of multiple isoenzymes of NO synthase (NOS) that catalyse the oxidation of L-arginine to L-citrulline in a nicotinamide adenine dinucleotide phosphate reduced form (NADPH) and oxygen-dependent reaction. The three isoforms of NOS are NOS1, NOS2, and NOS3, sharing a 51–57% homology. The product of the NOS1 gene is commonly known as neuronal NOS (nNOS), while that of NOS3 is known as endothelial NOS (eNOS). These isoforms are constitutive and calcium-dependent. The third isoform encoded by the gene NOS2 is an inducible NOS (iNOS) and represents the calcium-independent isoform [2]. iNOS is activated only by cytokines such as lipopolysaccharide, interleukin 1, and tumor necrosis factor alpha (TNFα) [3].

NO can also be formed independently of the action of NOS in the process of reduction of nitrite occurring spontaneously under hypoxic and/or acidic conditions or mediated by the action of different enzymes such as xanthine oxidase and cytochrome oxidase c [2]. Different from the conventional biosignaling pathway that provides, as a first step, the binding of a molecule to its receptor, the high solubility and reactivity are properties that allow NO to freely diffuse into biological membranes [1] and to exert its role with direct binding to biological macromolecules [1]. In the major signalling pathway, NO-mediated binding with soluble guanylate cyclase (sGC) occurs, which catalyses the conversion of guanosine-5′-triphosphate (GTP) to cyclic guanosine-3′,5′-monophosphate (cGMP). cGMP represents the main effector for NO activity, which transmits the NO signal to downstream elements, such as cGMP-dependent protein kinase, cGMP-gated cation channels, and cGMP-regulated phosphodiesterase [4]. In addition, NO shows high reactivity to the iron of the heme-containing enzymes (such as cyclooxygenase and cytochrome P450 steroidogenic enzymes), iron-sulphur groups, and thiol (-SH) group-containing proteins [5]. The capacity to act as a free radical scavenger is also attributable to NO, with particularly high reactivity for the superoxide anion (O_2_^−^) [5].

NO exerts relevant regulatory functions in a variety of physiological processes, including vasodilatation, neuronal signalling, and the immune response [5]. In detail, NO maintains a low vascular tone and prevents the adhesion of leukocytes and platelets to the vascular wall or acts as a mediator in some central neurons and peripheral nerve endings. More to the point, it plays a role in macrophages during the inflammatory response [6]. In addition, several studies suggested a function of NO as an autocrine/paracrine regulator in several aspects of the reproductive process [7]. In men, neurogenic NO has a role in the initiation and maintenance of penile erection [8] but also in the regulation of testicular functions such as spermatogenesis and steroidogenesis [9].

Concerning the female side, several authors focused on evaluating the role of nitric oxide within the ovary and, for this purpose, the aim of the present review is to describe the current literature about the involvement of nitric oxide in several aspects of ovarian physiology, and in the context of in-vitro fertilization (IVF).

## 2. Nitric Oxide Roles

### 2.1. NOS Expression in the Ovary

In the ovary, NO can be generated not only by ovarian cells, but also by the ovarian vasculature and by the resident or infiltrating macrophages [2,10]. Regarding the ovarian cells, several authors explored the expression and the localization of the NOS isoforms within the ovary, finding substantial divergences among the different species. In rats, eNOS is predominant in mural granulosa cells, the theca layer, ovarian stroma, and ovarian blood vessels [11], while iNOS is restricted to somatic cells of primary, secondary, and small antral follicles and luteal cells [11]. On the contrary, in mouse ovary, both eNOS and iNOS are expressed in theca and granulosa cells [11]. In the bovine ovary, eNOS is expressed in theca and granulosa cells, together with the surface epithelium and luteal cells [2]. In addition, in porcine ovary, eNOS is found more frequently than iNOS in granulosa cells [12]. Inducible NOS and eNOS have been shown to be expressed in human granulosa and luteal cells [7,13].

### 2.2. Nitric Oxide and Steroidogenesis

The experimental use of compounds functioning as NOS inhibitors or NO donors allows researchers to determine the role of nitric oxide in ovarian steroidogenesis, finding that NO represents a key regulator in this process. The involvement of nitric oxide in steroidogenesis is represented in Figure 1, and the studies analysing the role of nitric oxide in this process are summarized in Table 1.

It has been demonstrated in vitro the function of NO in inhibiting the estradiol/progesterone secretion in granulosa/luteal cells in rat [10,14], bovine [19], porcine [17], and human [13,18] models.

In the presence of an NO donor, such as sodium nitroprusside (SNP), the progesterone synthesis in cultures of rat ovarian dispersates was inhibited in a dose-dependent manner, with 10^−3^ M of SNP being the concentration with the maximum effect in the inhibition of progesterone synthesis [14]. In addition, in human granulosa/luteal cells cultivated in vitro, the administration of NO donors S-nitroso-L-acetyl penicillamine (SNAP) and S-nitroso-glutatione dose-dependently inhibited both estradiol and progesterone secretion, while the administration of selective NO synthase inhibitors significantly increased estradiol secretion [13].

In addition, researchers focused their interest in investigating the molecular mechanism at the base of this regulation and demonstrated that NO exerts its inhibitory effect on aromatase activity, a key enzyme in the steroidogenic pathway [17,18]. The culture of human granulosa cells added to NO donors SNAP and ethanamine 1,1-diethyl-2-hydroxy-2-nitrosohydrazine (NOC12) resulted in an inhibition of aromatase activity in a dose-dependent manner [18].

The aromatase inhibition is possible via two different mechanisms. The direct inhibitory effect on the enzyme is mediated by the formation of a nitrosothiol group in the cysteine residue of aromatase enzyme [18], while the indirect mechanism provides the downregulation of aromatase transcription, with decreasing mRNA values for the enzyme.

It has been shown that the inhibitory effect of nitric oxide on steroidogenesis may be partially mediated by two important prostaglandins (PGs) released in granulosa cells: PGE_2_ and PGE_2α_. For this purpose, Basini and colleagues (2001) [19] collected bovine granulosa cells from small (<5 mm) and large (>8 mm) follicles and treated these samples with SNAP or with indomethacin (an inhibitor of PGs synthesis). It was previously demonstrated that SNAP inhibits estradiol and progesterone production in bovine granulosa cells [20], and the experiments by Basini and colleagues (2001) suggest an involvement of PGE_2_ in determining the effect of SNAP on progesterone production. Similarly, the inhibitory effect induced by SNAP on estradiol production could be mediated by PGE_2_ in cells from large follicles and by PGE_2α_ in those from small follicles [19].

The experimental in vitro studies demonstrate that NO inhibits estradiol and progesterone production while NOS inhibitors enhance steroid synthesis by granulosa/luteal cells, but a study conducted using an ex-vivo rat ovary perfusion model showed different results in this context [15]. The authors demonstrated that the addition of the NOS inhibitor N(G)-monomethyl-L-arginine (L-NMMA) did not affect steroid concentration in the perfusion media, while the administration of NO donor spermine NONOate increased progesterone production. This is in accordance with the results of the experiments conducted on whole cultured rat ovaries in which an NO donor caused a dose-dependent increase in progesterone synthesis with a concomitant decrease in estradiol secretion [16].

Considering the differences existing between the experimental models, further investigation is needed to elucidate the role of nitric oxide in the steroidogenesis process.

### 2.3. Nitric Oxide and Folliculogenesis

Ovarian folliculogenesis is a dynamic process regulated by many factors in which the complex crosstalk between apoptotic cell death and cell growth signals causes a large number of follicles to be eliminated via atresia, and only few follicles have the chance to ovulate during the reproductive life. Follicular development induced by pregnant mare serum gonadotrophin (PMSG) in a rat model resulted in an increase in eNOS expression in granulosa cells [21,22]. The subsequent stimulation with human chorionic gonadotropin (hCG) induced an increased expression of both isoforms (eNOS and iNOS) [22], suggesting a role of nitric oxide in follicular development.

A line of research highlighted the double role of nitric oxide during the process of folliculogenesis, showing that, based on its concentration, nitric oxide may exert a role in protecting or inducing follicular apoptosis [23]. In detail, the NO donor SNAP (10^−3^, 10^−4^, 10^−5^ M) was added to bovine granulosa cells collected from follicles divided according to their size, small (<5 mm) and large (>8 mm). The highest concentration of the NO donor significantly inhibited DNA fragmentation in all the cells whereas the lowest concentration stimulated cellular apoptosis only in granulosa cells from large follicles [23]. However, several authors published data in favour of an antiapoptotic role of NO [24,25,26,27] by inhibiting Fas-FasL system-mediated apoptosis in rat and bovine granulosa cells cultured in vitro [25,27].

In detail, it has been demonstrated that the decrease in iNOS expression in rat granulosa cells is accompanied by the involvement of the Fas/FasL system in inducing apoptosis through the activation of a caspase-mediated cascade [25]. In addition, in a bovine model, the inhibition of iNOS expression increased FasL mRNA levels in granulosa cells cultured in vitro, resulting in activation of caspase-3 and, consequently, in an increased incidence of apoptotic cell death [27].

In addition, a role of the cytostatic factor in the granulosa cells from immature follicles has been attributed to NO [24,25,28], suggesting that the presence of iNOS is a requirement for immature follicles to stay quiescent [28] and that the alteration in iNOS expression in granulosa cells of immature follicles may represent a trigger for rendering them atretic or developing follicles [24,28]. To confirm this, there are data showing a decrease in iNOS mRNA levels in cultured rat granulosa cells after administrating two different agents driving immature follicles into different developmental stages such as EGF (involved in follicular development by suppressing granulosa cells apoptosis) and Buserelin (GnRH agonist that induces apoptosis and inhibits cell proliferation in granulosa cells) published by Matsumi et al., (2000) [28]. The studies presented in this paragraph are summarized in Table 2.

### 2.4. Nitric Oxide and Oocyte Meiotic Maturation

In the developing follicle, the fully-grown mammalian oocytes remain arrested at the prophase of meiosis I, at the stage of diplotene [29], where oocytes are distinguishable by the presence of a large nucleus, referred to as a germinal vesicle (GV). The luteinizing hormone (LH) surge during the mid-cycle promotes the meiotic resumption and the oocyte maturation, with the “germinal vesicle breakdown” (GVBD) characterized by the disappearance of the oocyte nucleus [30]. Oocyte maturation is a process characterized by active communication between granulosa cells and the oocyte gap junction-mediated process (in particular, connexin 43 and connexin 37). In response to the LH surge, the decrease in gap junction permeability and the interruption of intercellular communication within the follicle precedes the meiotic resumption and oocyte maturation [31].

Nitric oxide seems to exert a central role among the molecules involved in the modulation of the meiotic cell cycle in mammalian oocytes (Table 3). The reduced iNOS expression in granulosa cells and reduced NO levels in follicular fluid in a rat model during LH/hCG-induced meiotic resumption from diplotene arrest [32,33,34] strengthened the notion that the NO pathway plays an important role in oocyte meiotic maturation. In addition, eNOS knock-out mice exhibited an impairment in oocyte maturation, with fewer oocytes entering metaphase II and a higher percentage of oocytes at metaphase I or with atypical morphology compared to wild-type mice [35,36,37].

In this context, the level of nitric oxide is still a source of investigation since several studies conducted in vitro suggest that increased levels of iNOS/eNOS-induced NO in granulosa cells and cumulus cell-enclosed oocytes restart the oocyte meiotic progression from diplotene arrest [22,35,36,37,38,39,40,41], while other authors showed that a reduction in the NO level causes the resumption from diplotene arrest [42,43,44]. Other lines of research published data about the double role of NO (stimulatory or inhibitory) depending on its concentration using mouse [43,45] and bovine [46] cumulus-enclosed oocytes. In detail, high concentrations (1, 100, and 500 µM) of the NO donor (SNP) prevented GVBD in bovine cumulus-enclosed oocytes. On the contrary, at a lower concentration of 0.1 µM, SNP stimulated meiotic resumption [46]. In agreement with this data, Bu et al., 2003 [43] demonstrated that a low concentration of SNP (10^−5^ M) stimulated meiotic resumption in the presence of hypoxantine; on the contrary, a high concentration of SNP did not have any effects on GVBD but resulted in lower percentages of cumulus-enclosed oocytes reaching metaphase II stage [43]. The study published by Botigelli and colleagues (2017) established the role of SNAP in delaying meiosis resumption by the NO/cGMP pathway, by increasing cGMP [47].

It is noteworthy that the molecule cGMP is produced by the granulosa cells and is transported via gap-junctions into the oocytes, exerting a central role in the maintenance of oocyte meiotic arrest in preovulatory follicles. In addition, mitogen-activated protein kinase (MAPK) plays an important role in initiating oocyte maturation [32].

In an in vitro study using cultured rat ovaries, different roles of iNOS-induced and eNOS-induced NO after LH surge were proposed by Nakamura et al., 2002 [32]. The LH surge induces an increased expression of eNOS in the theca cell layer, favouring the blood supply to the preovulatory follicles, while iNOS expression decreases in granulosa cells, inducing the subsequent decrease of the intrafollicular NO concentration [32]. More in detail, before the LH surge, the high concentration of iNOS-induced NO induces the increase in intrafollicular cGMP and the maintenance of meiotic arrest [45]; on the contrary, the LH surge suppresses iNOS expression. The consequent drop of NO determines the decline in the level of intrafollicular cGMP. This allows the sequential activation of RAF, MAP2K1, and MAPK, with consequent disruption of gap junctional communication in the somatic follicular cells. The breakdown of gap junctional communication stops the transfer of the cAMP from the granulosa cells to oocytes, enabling the resumption of meiosis. The activation of MAPK also leads to cumulus expansion and ovulation in rat follicle-enclosed oocytes cultured in vitro [48]. The schematic view of the involvement of NO during the process of GVBD and meiotic resumption after LH surge/hCG administration is illustrated in Figure 2.

### 2.5. Nitric Oxide and IVF

Limited data are published concerning the role of NO during the process of embryo development and in the context of IVF. The information in this field indicates that the role of NO in preimplantation embryo development is not yet clear and that varying levels of NO may have different regulatory effects. The addition to sperm and oocyte media of the NOS inhibitor L-NAME at high concentrations (5, 10 mM) resulted in an inhibition of mouse embryo development, while the addition of low concentrations of the NO donor L-arginine stimulated the fertilization rate and embryonic development [49]. High concentrations of L-arginine also inhibit fertilization and embryo development processes [49,50], suggesting that an appropriate level of NO is crucial for the correct progression of these processes, as concluded in a previous report published by Barroso et al., 1998 [51] where higher concentrations of NO inhibited embryo development in vitro and implantation in vivo in mice. Indeed, the addition of 0.25 mM L-NAME to embryo culture medium significantly reduced the proportion of mouse embryos developing beyond the two-cell stage [52], while the addition of 0.5 mM L-NAME was found to compromise the percentage of embryos that cavitated [53].

Moreover, cGMP seems to be the molecular effector of NO in preimplantation embryo development [51,54]. The use of the NO inhibitor NG-nitro-L-arginine (L-NA) in a mouse embryo in vitro culture system blocked the development of two-cell embryos to the four-cell stage but this inhibitory effect was lost when the concentration of L-NA decreased to 125 µM. The administration of the NO donor SNP at all concentrations analysed (ranging from 0.2 nM to 500 µM) inhibited normal embryo development, but the combination of L-NA and SNP reversed these inhibitory effects. The fact that NO regulates embryo development via the cGMP pathway was demonstrated by adding to the culture medium both 8-Br-cGMP (derivative of cGMP) and 500 µM L-NA. Embryos showed a significant increase in normal development when compared with those cultured in 500 µM L-NA alone. The inhibitory effect of L-NA was reversed by 8-Br-cGMP, indicating a role of the NO/cGMP pathway in preimplantation embryo development [51].

The studies from the human side, in this context, showed contradictory results. On the one side, Battaglia et al. (2003) [55] and Lipari et al. (2009) [56] found an association between the nitric oxide metabolites released in the embryo culture media with blastocyst formation progression [55] together with the pregnancy rates [56]. In detail, the mean NO metabolites in the insemination media were 2.6 higher in embryos that progressed to the blastocyst stage on day 5 of culture compared with those that did not [55], while the NO metabolite concentrations of the culture medium of each embryo were higher in the pregnant-resulting patients than those who did not become pregnant [56]. On the contrary, no association was found between NO metabolite production in a culture medium with the embryo cleavage rate in the study of Gallinelli et al., 2009 [57]. In addition, the administration of an NO donor, such as nitroglycerin (NTG), to IVF patients with a history of implantation failure was no more effective in improving implantation or pregnancy rates compared to a placebo drug [58].

More to the point, other authors who conducted clinical studies investigated the possible association between serum/follicular fluid NO concentration and oocyte/embryo quality. Few data have been published and, in addition, are also conflicting.

The lack of a correlation between variations in the follicular fluid concentrations of NO and tumor necrosis factor alpha (TNFα) with oocyte quality and in vitro fertilization/embryo transfer (IVF-ET) parameters was shown in the study of Lee et al. (2000) [59]. A few years later, it was shown that higher follicular levels of nitrite and nitrate were found in those follicles resulting in poor-quality embryos [60]. These data are in accordance with those published by Vignini et al., (2008) [61] where the mean concentration of NO in follicular fluid was higher in patients with poor-quality embryos than those with high-quality embryos (57.54 ± 12.67 nmol/mg versus 42.43 ± 16.32 nmol/mg) in a statistically significant manner [61]. A relatively recent study investigated the possible role of iNOS expression in cumulus cells (CCs), finding that the amount of iNOS and heme oxygenase (HO-I) mRNA and protein resulted higher in CCs belonging to oocytes that were not fertilized, compared to CCs corresponding to oocytes that were normally fertilized [62]. These data suggest a role of iNOS as an oocyte biomarker, but further investigation is needed to consider this biomarker a useful tool for oocyte selection. Moreover, additional data are necessary to elucidate the possible correlation between NO and embryo development and IVF parameters.

The experimental and clinical studies on the role of NO in the context of IVF are summarized in Table 4 and Table 5, respectively.

## 3. Discussion

The present work reviews the current studies regarding the implication of NO in several aspect of ovarian physiology. NO is generated by ovarian cells as demonstrated by several studies examining the pattern of NOS isoforms in ovaries of different species [11,12,13]. It is well known that the nitrergic system has a central role in vascular endothelium, in central, and peripheral neurons, together with macrophages, and in immunity and inflammatory systems. Considering that neurons, blood vessels, macrophages, and immunity cells are integral parts of the ovary, it is not unlikely that the production of NO in the ovary also derives from these other cell lines.

Several studies have demonstrated the role of NO within the ovary. First of all, the regulation of steroidogenesis in granulosa/luteal cells in different species seems to be clarified [10,13,14,15,16,17], together with the molecular mechanism at the base, which involves the direct binding to P450 aromatase or its transcriptional inhibition [16,17] or the inhibition of androstenedione secretion [63]. Another topic deeply investigated is the association between NO and oocyte competence, with particular regard for the role of NO in the resumption of meiotic progression from the prophase of the first meiotic division up to the metaphase II stage. On the one hand, it is clear that NO is involved in this process, but on the other, controversial data exist regarding the intracellular level of nitric oxide in granulosa cells/cumulus-enclosed oocytes that can induce the resumption of meiotic progression. Concerning folliculogenesis, it is known that is a process involving two important mechanisms: cell growth and apoptosis leading to follicular development and atresia, respectively. While a role of NO has been elucidated in the protection from apoptosis and as a cytostatic factor in the granulosa cells of immature follicles, the exact function of NO in the process of cell growth is yet to be elucidated.

Another important aspect to highlight is the fact that most of the studies published in this field were carried out on experimental in vitro or in vivo animal models. There are not many studies investigating the function of NO within human granulosa/luteal cells or oocytes [7,13,17,64]. In this context, information has been published about the role of NO as an autocrine regulator of human granulosa/luteal cell steroidogenesis [13,17] and on the proapoptotic effect of NO in the human corpus luteum [7]. In addition, insights into the function of NO in the regulation of ovulation have been elucidated in a study conducted on human granulosa cells by Fang et al., 2015 [65]. Controversial results have been published concerning the function of nitric oxide in the process of human embryo development. If some authors showed a positive role of NO in developing embryos [55,56], others failed to find the same results [57,59,60,61]. In addition, the possible association between serum/follicular fluid nitric oxide concentration and embryo quality together with IVF outcomes has been investigated [59,60,61,62], but additional studies are needed for a more comprehensive overview in this regard.

In the literature, the role of NO has also been investigated in other contexts involving the ovary.

First of all, the role of free radicals related to NO has been established, but further data are needed in this perspective. For example, peroxynitrite (ONOO^−^) that origins from NO and anion superoxide (O_2_^•−^) affects the viability of cumulus cells [66] and the oocyte spindle structure in mouse models and in a dose-dependent manner [66,67].

The multifaceted association between NO and the ovarian cancer biology has also been investigated. Apoptosis and survival of ovarian cancer cells, interaction of stromal and ovarian cancer cells, angiogenesis, and chemioresistance of ovarian cells seemed to be the mechanisms most regulated by NO [68]. In addition, the dose of NO together with the species of synthesis enzymes and the surrounding microenvironment are important aspects that influence NO signaling in ovarian cancer, but remain to be investigated in detail [68].

Few data have been published regarding the relationship between NO and ovarian cysts. The existing studies concentrated on the association between NO and endometriotic cysts in order to deepen the knowledge of the complex mechanisms of angiogenesis that have been implicated in the pathogenesis of endometriosis [69] and in evaluating the association between NO and PCOS [70].

Menopause represents another phenomenon further investigated in relation to NO. The NO concentration in serum of women was altered in menopausal status [71,72,73], as well as in post-menopausal women with the metabolic syndrome [74], but further studies are needed to fully investigate this correlation.

In addition, literature data demonstrated that beneficial health effects may be due to dietary nitrate intake from the daily consumption of nitrate-rich foods. Vegetables and beetroot juice represent important sources of NO, since they contain a high amount of nitrate [75], which is associated with the positive effects of these food groups against type-2 diabetes and cardiovascular diseases [76,77] and in blood pressure lowering and in vasoprotective effects [78]. It would be interesting to investigate the possible effects of nitrate-rich food intake on the mechanisms regulating ovarian physiology.

## 4. Conclusions

It is noteworthy that NO exerts remarkable functions within the ovary, including the control of steroidogenesis, folliculogenesis, and the oocyte competence but the precise mechanisms by which it carries out its effects need further investigation, in particular in the context of human ovarian physiology.

## Figures and Tables

**Figure 1 ijerph-18-00980-f001:**
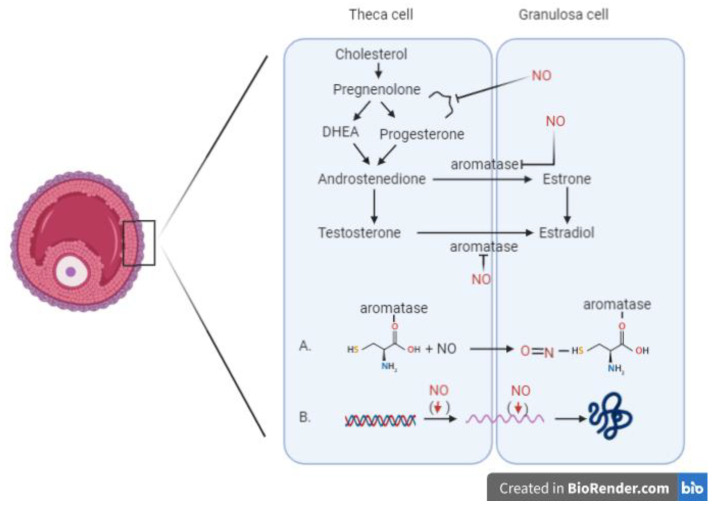
Involvement of nitric oxide in the process of steroidogenesis. Two different mechanisms inhibiting the aromatase enzyme: (**A**) Direct inhibitory effect of NO mediated by the formation of a nitrosothiol group in the cysteine residue of the aromatase enzyme. (**B**) Indirect mechanism resulting in the downregulation of aromatase transcription, with decreasing mRNA values for the enzyme. (DHEA: Dehydroepiandrosterone)

**Figure 2 ijerph-18-00980-f002:**
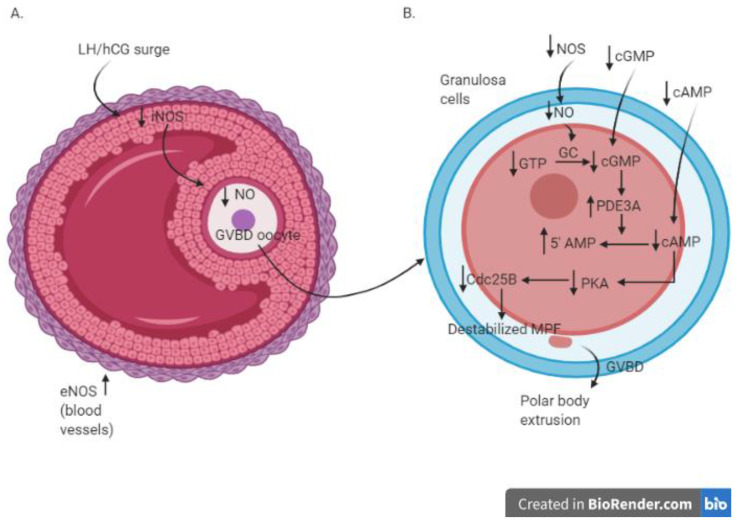
Molecular cascade in antral follicle (**A**) and oocytes (**B**) during the process of GVBD and meiotic resumption after LH surge/hCG administration; AMP—3′-5′-cyclic adenosine monophosphate; cAMP—3′-5′-cyclic adenosine monophosphate; cGMP—cyclic guanosine monophosphate; Cdc25B—cell division cycle 25B; eNOS—endothelial nitric oxide sinthase; GC—guanylate cyclase; GVBD—germinal vescicle breakdown; hCG—human chorionic gonadotropin; LH—luteinizing hormone; MPF—M phase-promoting factor—NO: nitric oxide; NOS—nitric oxide synthase; PD3A—phosphodiesterase 3A; PKA—protein kinase A.

**Table 1 ijerph-18-00980-t001:** NO and steroidogenesis.

Authors	Animal	Model	NO Donors Used	NOS Inhibitors Used	Effects
Dave et al., 1997 [10]	Rat	In-vitro granulosa/lutein cells culture	-SNP: 10^−3^ mol/L, 10^−4^ mol/L -SNAP: 10^−3^ mol/L, 10^−4^ mol/L		-Progesterone synthesis inhibition in the presence of the nitric oxide donors in a concentration-dependent manner.
Ahsan et al., 1997 [14]	Rat	In-vitro culture of dispersed ovarian cells	-SNP: 10^−4^ M and 10^−3^ M		-Progesterone synthesis in cultures of ovarian dispersates dose dependently inhibited by SNP and inversely related to the concentrations of nitrites measured in the culture medium. -Progesterone production stimulated by PGE_2_ (5 mM).
Mitsube et al., 1999 [15]	Rat	Ex-vivo perfused pre-ovulatory ovary	-Spermine NONOate: 10 µmol/L and 100 µmol/L	-L-NMMA: 300 µmol/L -AG: 300 µmol/L and 1 mmol/L	-Steroid concentration in the perfusion media not affected by the NO inhibitors. -NONOate (100 µmol/L) increased progesterone production.
Dong et al., 1999 [16]	Rat	In-vitro cultured ovaries	-DETA/NO: 10^–6^, 10^–5^, 10^–4^ M -DETA: 10^–4^M	-L-NAME: 10^–4^ M	-Dose-dependent increase in progesterone synthesis, with a concomitant decrease in ovarian oestradiol secretion caused by DETA/NO. -No effects on progesterone and oestradiol secretion by L-NAME (10^−4^ M).
Masuda et al., 2001 [17]	Porcine	In-vitro granulosa cells derived from small-sized (<3 mm) and medium-sized (3–5 mm) ovarian follicles	-NOC18: 0.01–1.0 mmol/L	-LNMMA: 0.01–1.0 mmol/L	-NOC18 suppressed the basal and gonadotropin-stimulated E_2_ and inhibited progesterone release from small-sized follicles. -No significant effect on progesterone release from medium-sized follicles by NOC18. -Significant release of E_2_ and progesterone from both small-sized and medium-sized follicles in the presence of gonadotropin by LNMMA.
Van-Voorhis et al., 1994 [13]	Human	In-vitro granulosa-luteal cell	-SNAP -S-nitroso glutathione	-LNMMA -N-nitro-arginase methyl ester.	-Inhibitors of NO synthase significantly increased estradiol secretion by granulosa-luteal cells. -NO donors caused a dose-dependent decrease in both estradiol and progesterone secretion.
Kagabu et al., 1999 [18]	Human	In-vitro ovarian granulosa-luteal cells	-SNAP: 10^−5^ M, 10^−4^ M, 10^−3^ M -NOC12: 10^−4^ M, 10^−3^M		-Aromatase activity significantly inhibited by treatment with SNAP (10^−4^ M, 10^−3^ M) and NOC12 (10^−4^ M, 10^−3^ M) in a dose-dependent manner. -Treatment with SNAP at 10^−3^ M decreased relative aromatase mRNA values and intracellular cyclic AMP concentrations.

AG—Aminoguanidine bicarbonate; DETA/NO—Diethylenetriamine/nitric oxide; DETA—Diethylenetriamine; L-NAME—NG-nitro-l-arginine methyl ester; L-NMMA—N(G)-monomethyl-L-arginine; NO—Nitric oxide; NOC12—ethanamine 1,1-diethyl-2-hydroxy-2-nitrosohydrazine; NOC18—1-Hydroxy-2-oxo-3,3-bis(2-aminoethyl)-1-trazene; NOS—Nitric oxide synthase; SNAP—S-nitroso-N-acetyl-penicillamine; SNP—Sodium nitroprusside; PGE2—Prostaglandin E2.

**Table 2 ijerph-18-00980-t002:** NO and folliculogenesis.

Authors	Animal	Model	NO Donors Used	NOS Inhibitors Used	Effects
Van Voorhish et al., 1995 [21]	Rat	In vivo			-Highest levels of iNOS mRNA in unstimulated ovaries. -After gonadotropin injection, iNOS mRNA declined to undetectable levels in ovaries containing ovulatory follicles before increasing slightly in ovaries containing copora lutea. -iNOS expressed in granulosa cells of secondary follicles and small antral follicles. -eNOS mRNA levels increased after gonadotropin stimulation and peaked in ovaries containing ovulatory follicles before declining in the luteal phase.
Jablonka-Shariff et al., 1997 [22]	Rat	In vivo			-In immature ovaries and during follicular development, iNOS was expressed in the theca cell layer and stroma. -After ovulation, iNOS was present only in the external layers of the developing corpus luteum.
Basini et al., 1998 [23]	Bovine	In vitro culture of granulosa cells collected from follicles divided according to their size, small (<5 mm) and large (>8 mm).	-SNAP: 10^−3^ M, 10^−4^ M, 10^−5^M		-Significant inhibition of DNA fragmentation in all the cells with the highest concentration of SNAP. -Stimulation of cellular apoptosis in granulosa cells from large follicles with the lowest concentration of SNAP.
Matsumi et al., 1998 [24]	Rat	In vitro culture of granulosa cells	-SNAP: 0.05 mM, 0.5 mM		-Inverse correlation between iNOS expression and apoptotic changes in rat granulosa cells. -The NO donor directly inhibited spontaneously occurring apoptosis.
Chen et al., 2005 [25]	Rat	In vitro culture of granulosa cells	-SNAP: 0.05 mM, 0.2 mM, 0.5 mM		-The induction of apoptosis in granulosa cells by 100 ng/ml rFasL in the presence of interferon-gamma was blocked by the concomitant addition of SNAP in a dose-dependent manner. -rFasL significantly up-regulated caspase-3, -8, and -9 activities in granulosa cells, which were attenuated by concurrent treatment with SNAP (0.5 mM).
Dineva et al., 2008 [26]	Human	In vitro human granulosa/luteinized cells	-SNP: 0.5 mM	-AG: 0.5 mM	-SNP treatment significantly lowered the caspase-3 activity that significantly increased after AG treatment.
Matsumi et al., 2000 [28]	Rat	In vitro culture of granulosa cells	-SNAP: 0.05 mM, 0.2 mM, 0.5 mM		-The in vitro induction of granulosa cell apoptosis by 10^−6^ M buserelin was inhibited by the addition of SNAP (0.5 mM) -The percentage of apoptotic cells increased by the addition of 10^−6^ M buserelin but this increase was reverted by supplementation of SNAP (0.5 mM) -Dose-dependent inhibition of DNA synthesis mediated by SNAP. At 0.5 mM, SNAP caused a 90% inhibition in DNA synthesis.

AG—Aminoguanidine bicarbonate; NO—Nitric oxide; NOS—Nitric oxide synthase; iNOS—inducible nitric oxide synthase; eNOS—endothelial nitric oxide synthase; rFasL—ecombinant Fas ligand; SNAP—S-nitroso-N-acetyl-penicillamine; SNP—Sodium nitroprusside; mRNA—Messenger RNA; DNA—deoxyribonucleic acid.

**Table 3 ijerph-18-00980-t003:** NO and oocyte maturation.

Authors	Animal	Model	NO Donors Used	NOS Inhibitors Used	Effects
Nakamura et al., 2002 [32]	Rat	In-vitro cultured ovaries and denuded oocytes	-SNAP: 500 μM	-AG; 0, 1 mM, 10 mM, 100 mM	-The percentage of oocytes at the germinal vesicle stage decreased in the group receiving 10 and 100 mM AG compared with the control group after 5 h of incubation. This GVBD-promoting effect of 100 mM AG was significantly reversed by the addition of 500 μM SNAP. -SNAP dose-dependently inhibited GVBD in denuded oocytes, and this effect of SNAP was reversed by the addition of hemoglobin
Yamagata et al., 2002 [34]	Rat	In vitro culture of granulosa cell and theca cell layers	-SNAP: 10 ^−5^ M, 10 ^−4^ M, 5 × 10^−4^ M		-Significant decrease in nitrate/nitrite concentration in the follicular fluid 5 and 10 h after hCG injection. -Significant decrease in iNOS mRNA expression 5 and 10 h after hCG injection. eNOS mRNA expression increased 5 and 10 h after hCG injection. -Decreased progesterone production and increased DNA fragmentation of granulosa cells with SNAP (10^−4^ M or 5 × 10^−4^ M).
Jablonka-Shariff et al., 1998 [35]	Mouse	In vivo (mice knock-out for the eNOS gene)			-eNOS knock-out females showed a significant reduction in ovulatory efficiency compared with wild type females. -eNOS deficiency impaired not only ovulation, but also oocyte meiotic maturation. -Fewer oocytes from eNOS knock-out mice entered metaphase II of meiosis, with a greater percentage in metaphase I or morphologically atypical relative to those in wild type mice.
Jablonka-Shariff et al., 1999 [36]	Rat	In-vivo		-L-NAME: 100 mg/kg administered orally -L-NIL, 30 mg/kg administered orally	-Fewer ovulated oocytes at metaphase II obtained from rats treated with NOS inhibitors, and a significantly greater percentage of oocytes displayed atypical morphology as compared with control oocytes.
Jablonka-Shariff et al., 2000 [37]	Mouse	In-vitro cultured COCs		-L-NAME: 0.1 mM, 0.5 mM, or 1 mM	-eNOS-KO mice contained fewer COCs relative to wild type mice. Maturation of COCs from eNOS-KO mice or wild type oocytes treated with L-NAME resulted in a lower percentage of oocytes at M II stage and a higher percentage of oocytes at M I or atypical stages compared with those from WT. - Abnormalities in the distribution of maternal chromosomes in MII stage-oocytes.
Sengoku et al., 2001 [38]	Mouse	In-vitro culture of oocytes/embryos	-SNP: ranging from 10^−3^ M to 10^−7^ M ÿ	-L-NAME: ranging from 10^−3^ M to 10^−7^ M	-Low concentrations of SNP (10^−7^ M) significantly stimulated meiotic maturation to metaphase II stages in CEOs. -L-NAME (10^−3^ M and 10^−5^ M) demonstrated a significant suppression in the resumption of meiosis. This inhibition was reversed by the addition of SNP. -No development beyond the four-cell stage was observed by the addition of a high concentration of SNP (10^−3^ M). -Inhibition of embryo development, especially the conversion of morulae to blastocysts, was also observed in the treatment with lower doses of SNP (10^−5^ and 10^−7^ M). -Inhibition of NO by an NOS inhibitor resulted in the dose-dependent inhibition of embryo development and hatching rates, but the concomitant addition of SNP with L-NAME reversed the inhibitory effect by each SNP or L-NAME treatment. -A low concentration of SNP (10^−7^ M) but not a high concentration of SNP (10^−3^ M) significantly stimulated trophoblast outgrowth, whereas the addition of L-NAME suppressed the spreading of blastocysts in a dose-dependent manner
Tao et al., 2004 [39]	Porcine	In-vitro cultured pre-antral follicles, COCs aspirated from medium follicles (3–6 mm in diameter), ovarian tissues, CL, CA and COCs from small (1–2 mm in diameter), medium (3–6 mm), and large follicles (7–10 mm)	-SNP: 0.1 M, 0.3 M, 0.5 M or 1 mM	-AG: 10 mM -L-NAME: 1 mM	-0.3, 0.5 or 1 mM SNP significantly inhibited antrum formation. -AG markedly inhibited porcine oocyte meiotic resumption while L-NAME inhibited first polar body (PB1) extrusion. -Porcine ovaries had distinct cell-specific expression of both eNOS and iNOS.
Tao et al., 2005 [40]	Porcine	In- vitro cultured CEOs and DOs.		-AG -L-NNA -L-NAME	-AG suppressed cumulus expansion and inhibited CEOs to resume meiosis but did not inhibit cumulus cell DNA fragmentation. -LNNA and L-NAME delayed cumulus expansion, inhibited cumulus cell DNA fragmentation, and inhibited CEOs to resume meiosis. -No effects on DO.
Huo et al., 2005 [41]	Mouse	In vitro culture of denuded and cumulus-enclosed GV-intact oocytes		-AG, 0, 1 mM, 10 mM, 50 mM	-AG significantly blocked the GVBD of the DOs in a dose-dependent manner. -AG blocked the PB1 emission of oocytes in a dose-dependent manner.
Pandey et al., 2015 [42]	Rat	In-vitro cultured oocytes	-SNAP: 0.1 mmol/L, 0.2 mmol/L, 0.3 mmol/L, and 0.4 mmol/L	-AG 0.0, 2.5 µmol/L, 5.0 µmol/L, 10.0 µmol/L, 20.0 µmol/L, and 40.0 µmol/L	-SNAP inhibited the spontaneous resumption of meiosis from diplotene arrest in a concentration-dependent manner. -AG induced the spontaneous resumption of meiosis in a concentration-dependent manner. -AG treatment resulted in a significant reduction in the level of iNOS expression in oocytes undergoing resumption of meiosis after 3 h of in vitro culture.
Bu et al., 2003 [43]	Mouse	In-vitro cultured CEOs and DOs	-SNP: ranging from 0.1-4 mM	-L-NAME: ranging from 10^−5^ M to 10^−2^ M -L-NNA: ranging from 10^−5^ M to 10^−2^ M.	-Low concentrations of SNP (10^−7^, 10^−6^, 10^−5^ M) significantly stimulated the oocyte meiotic maturation of CEOs but had no effect on DOs. -High concentrations of SNP (0.1 mM–4 mM) in the CEOs cultured in maturation medium resulted in a lower percentage of oocytes at the PB1 stage and a higher percentage of atypical oocytes in a dose-dependent manner compared with the control. -Treatment with SNP (1 mM) resulted in a significant inhibitory effect on the formation of PB1. -The concentration of SNP (1 mM) significantly delayed GVBD during the first 5 h of the culture period. The concomitant addition of L-NAME with SNP did not reverse the inhibitory effect of SNP on CEOs.
Tripathi et al., 2009 [44]	Rat	In-vitro culture of oocytes			-Tonic level of H_2_O_2_ induced the meiotic resumption of diplotene-arrested oocytes and a further increase may have led to apoptosis. -Reduction in iNOS expression and total nitrite level associated with meiotic resumption in diplotene-arrested oocytes, but induced apoptosis in aged oocytes.
Bu et al., 2004 [45]	Mouse	In-vitro cultured CEOs	-SNP: 1 mM, 10 µM.		-SNP (1 mM) significantly delayed GVBD during the first 5 h of the incubation period -SNP (1 mM) inhibited the formation of PB1 at the end of 24 h of incubation. -SNP (10µM) stimulated the meiotic maturation of oocytes by overcoming the inhibition of hypoxantine.
Groessels et al., 2007 [46]	Bovine	In vitro cultured CEOs	-SNP: 100 µM, 500 µM	-AG: 1 mM,10 mM and 50 mM	-AG (10 and 50 mM) and SNP (100 and 500 mM) significantly inhibited GVBD after 7 h of culture. -SNP (0.01 mM) stimulated GVBD.

AG—Aminoguanidine bicarbonate; CA—Corpus albican; CEOs—Cumulus-enclosed oocytes; CL—Corpus luteum; COCs—Cumulus-oocyte complexes; DOs—Denuded oocytes; eNOS—endotelial nitric oxide synthase; GVBD—Germinal vesical breakdown; hCG—human chorionic gonadotropin; H_2_O_2_—hydrogen peroxide; iNOS—inducible nitric oxide synthase; L-NAME—NG-nitro-l-arginine methyl ester; L-NIL—L-N6-(1-iminoethyl)-lysine; L-NNA—Nomega-nitro-L-arginine; PB—Polar body; KO—knock-out; NO—Nitric oxide; NOS—Nitric oxide synthase; SNAP—S-nitroso-N-acetyl-penicillamine; SNP—Sodium nitroprusside.

**Table 4 ijerph-18-00980-t004:** Experimental studies investigating the role of NO in IVF.

Authors	Model	NO Donors Used	NOS Inhibitors Used	Effects
Kim et al., 2004 [49]	Mice (In vivo)	-L-NAME (0.5, 1, 5 and 10 mM)		-L-NAME inhibited the fertilization rate and the early embryonic development by treating sperm or oocytes. -Fertilization rate and early embryonic development were reduced when L-NAME or L-arginine was added to the culture media of embryos. -Microinjection of L-NAME into the fertilized embryos inhibited in a dose-dependent manner early embryonic development, but only by high concentrations of L-arginine.
Santana et al., 2014 [50]	Bovine (in vitro embryo culture)	-L-NAME (10 mM) -L-arginine (1 mM, 10 mM, or 50 mM)		-Supplementation with L-NAME from Day 1 to 8 of the culture decreased blastocyst and hatching rates. -L-arginine (50 mM) added from Day 1 to Day 8 decreased the blastocyst rates; in contrast, when added from Day 5 to 8, L-arginine (1 mM) improved the embryo hatching rates and quality. -Positive correlation between NO levels in the medium during this culture period and increased embryo hatching rates and quality.
Barroso et al., 1998 [51]	Mice (In vitro embryo culture and in vivo)	-DETA/NO: 0.001 mM, 0.01 mM, 0.1 mM, 1mM -DETA: 0.1 mM, 1 mM In-vivo: S.c. implantation of miniosmotic pumps containing either saline or different concentrations of DETA/NO or DETA (0.2, 0.4, and 0.8 mM) to deliver a daily dose of 5, 10 or 20 µmol per animal.		-None of the embryos progressed beyond the 4-cell stage when exposed to DETA/NO (0.1 or 1.0 mM). -Embryo development unaffected by lower (0.001 and 0.01 mM) concentrations of DETA/NO, after 48 h preincubation with DETA/NO or DETA only. -Embryo implantation inhibition with the infusion of DETA/NO in a dose-dependent manner. -No implantation sites observed with the infusion of a daily dose of 20 µmol DETA/NO, compared with the control or DETA-treated mice.
Tranguch et al., 2003 [52]	Mice (In vitro embryo culture)	-SNP: from 0.1 to 500 mM	-l-NA: from 125 to 500 mM	-All three NOS isoforms were expressed in two-cell, four-cell, morula, and blastocyst embryos. -Blastocyst-stage embryos isolated on the midmorning of Day 4 of pregnancy expressed only nNOS and eNOS, whereas those isolated in the midafternoon expressed all three NOS isoforms.
Gouge et al., 1998 [53]	Mice (in vitro embryo culture)		-l-NA: 500 µM	-Preimplantation murine embryos produced NO, reversibly inhibited by the culture of embryos in medium containing L-NA. -L-NA inhibits normal embryo development.
Chen et al., 2001 [54]	Mice (in vitro embryo culture)	-SNP: 0.1 µM, 1 µM, 10 µM	-L-NAME: 0.1 µM, 1 µM, 10 µM	-L-NAME inhibited blastocyst development in a concentration-dependent manner and SNP (0.1 µM) reversed this effect. -Excessive NO (> or = 10 µM) induced apoptosis in the mouse embryos. -The inhibitory effect of L-NAME was reversed by 8-Br-cGMP that rescued the embryo growth. -ODQ inhibited the embryo development in a dose-dependent manner (0.1 µM–100 µM) but had no effect on NO-induced embryo apoptosis.

8-Br-cGMP—8-Bromoguanosine 3′,5′-cyclic monophosphate; DETA/NO—Diethylenetriamine/nitric oxide; DETA—Diethylenetriamine; IVF—In-vitro fertilization; L-NAME—NG-nitro-l-arginine methyl ester; l-NA—NG-nitro-l-arginine; NO—Nitric oxide; NOS—Nitric oxide synthase; ODQ—1H-[1,2,4]oxadiazolo[4,3-a]quinoxalin-1-one; S.C—Subcutaneous; SNP—Sodium nitroprusside.

**Table 5 ijerph-18-00980-t005:** Clinical studies investigating the role of NO in IVF.

Authors	Type of Study	Population Considered	Effects
Battaglia et al., 2003 [55]	Observational (pilot) study	-23 women undergoing an IVF-ET program	-Embryonic secretion of NO demonstrated by higher mean nitrite/nitrate concentrations present in the culture medium of each embryo -The mean nitrite/nitrate concentrations in the embryo culture medium were significantly higher in patients that became pregnant.
Lipari et al., 2009 [56]	Observational study	-11 women undergoing an IVF-ET program	-Nitric oxide metabolite levels in the insemination media were higher in embryos that progressed to blastocysts by culture day 5 compared with those that did not.
Gallinelli et al., 2008 [57]	Study population	-179 women undergoing an IVF-ET program: -123 women with fresh oocytes. -56 oocyte thawing cycles	-Higher NO production in embryos derived from ICSI than from IVF after 52 h of culture. -Embryos derived from fresh oocytes produced more NO than embryos from thawed oocytes after 48 and 52 h of culture.
Ohl et al., 2002 [58]	Prospective, double-blind, randomized, placebo-controlled trial	-138 patients undergoing an IVF-ET program: -70 patients treated with nitroglycerin-68 with placebo	-No improved implantation or pregnancy rates after NTG treatment on the day before embryo transfer.
Lee et al., 2000 [59]	Observational study	-43 patients undergoing an IVF-ET program	-No significant correlation between the concentrations of NO and TNF-alpha in follicular fluid. -NO levels in follicular fluid were altered in infertility-associated diseases. -TNF-alpha levels but not NO levels influenced oocyte quality.
Lee et al., 2004 [60]	Prospective, case-control study	-36 patients undergoing an IVF-ET program: -18 patients undergoing an IVF-ET program with tubal or peritoneal factor -18 female partners from couples with male factor infertility	-Higher follicular NO levels associated with advanced fragmentation of embryos. -Higher serum NO levels found among non-pregnant patients with tubal or peritoneal factor infertility.
Vignini et al., 2008 [61]	Observational study	-15 patients undergoing an IVF-ET program	-Mean concentration of the NO follicular fluid was significantly higher in patients with embryos showing significant or severe fragmentation or blastomeres of distinctively unequal size than those with good-quality embryos. -Direct correlation between follicular NO and embryo grading and an inverse correlation between follicular NO and serum 17beta-estradiol.
Bergandi et al., 2014 [62]	Observational study	-40 women undergoing an IVF-ET program	-iNOS and HO-1 mRNAs and proteins significantly higher in cumulus cells (CCs) corresponding to oocytes that were not fertilized in comparison to CCs whose corresponding oocytes showed normal fertilization.

CCs—Cumulus cells; ET—embryo transfer; HO-1—Heme oxygenase 1; ICSI—intracitoplasmic sperm injection; iNOS—inducible nitric oxide synthase; IVF—In-vitro fertilization; NO—Nitric oxide; NTG—Nitroglycerin; TNF-alpha—Tumor necrosis factor alpha.

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
