# Peer review of "Novel Insights on the Role of Nitric Oxide in the Ovary: A Review of the Literature"

_ijerph, 2021, doi:10.3390/ijerph18030980_

Round 1

Reviewer 1 Report

The topic of the review is very interesting as authors state NO exerts important function on overall ovary physiology. Literature has been exhaustively reviewed and results are clearly detailed in a large number of tables (5) and figures (2).

I only have some minor concerns:

  • Introduction:

I miss a more detailed explanation or revision about free radical biology related to NO. For instance, peroxynitrite (ONOO−) origins from NO and superoxide (O2•−), and it is a highly reactive ion responsible for macromolecular damage. Is there any work related to ovarian physiology where peroxynitrite damage could be in part responsible for disruptive functions? If there are many, you could consider to discuss a little bit this topic on the review.

  • Line 33: consider to put the gene names in italics
  • Line 170: perhaps the last sentence of the paragraph needs a reference.

Author Response

Dear Editor,

Thank you for the reviewers comments concerning our manuscript entitled “Novel insights on the role of nitric oxide in the ovary: a review of the literature“. Comments were very helpful for improving the manuscript. The revised portions in the text and the added references are highlighted in yellow. The point-by-point responses to the Reviewer comments are as follow:    

Reviewer 1

Q1.I miss a more detailed explanation or revision about free radical biology related to NO. For instance, peroxynitrite (ONOO−) origins from NO and superoxide (O2•−), and it is a highly reactive ion responsible for macromolecular damage. Is there any work related to ovarian physiology where peroxynitrite damage could be in part responsible for disruptive functions? If there are many, you could consider to discuss a little bit this topic on the review.”

A1. Thanks for the interesting suggestion. Peroxinitrite is not widely discussed in the literature but the existing data demonstrated that peroxynitrite “affects the viability of cumulus cells (Banarijee et al., 2013) and the oocyte spindle structure in mouse models and in a dose-dependent manner (Banarijee et al., 2013; Khan et al., 2016)” (Lines 362-363, References 66,67).

Q2. Line 170: perhaps the last sentence of the paragraph needs a reference.

A2. We added the lacking reference at the end of the sentence (Reference 27).

Reviewer 2.

Q1.Is there any relationship shown for the role of NO, present in the ovary, with the ovarian cancer?” 

A1. Thanks to the reviewer for the interesting questions and suggestions for future research.

The role of NO in ovarian cancer has been studied and a recent review published in 2016 (El-Sehemy et al., 2016) analyzed “the multifaceted association between nitric oxide and ovarian cancer biology. Apoptosis and survival of ovarian cancer cells, interaction of stromal and ovarian cancer cells, angiogenesis and chemioresistance of ovarian cells seemed to be the mechanisms most regulated by NO. In addition, the dose of NO together with the species of synthesis enzymes and the surrounding microenvironment are important aspects that influence the NO signaling in ovarian cancer, yet to be investigated in detail” (Lines 364-370, Reference 68).

Q2. “Any relationship to ovarian cyst?” 

A2. “Few data have been published regarding the relationship between NO and ovarian cysts. The existing studies concentrated on the correlation between NO in endometriotic cysts in order to deepen the knowledge of the complex mechanisms of angiogenesis that has been implicated in the pathogenesis of the endometriosis (Goteri et al., 2010) and in evaluating the association between NO and PCOS (Meng et al., 2019)” (Lines 371-375, References 69,70).

Q3. “Beneficial and negative effects of NO in old vs young women OR pre-menopause vs post menopause?”

A3. Nitric oxide concentration in serum of women resulted altered in menopausal status (Watanabe et al., 2000; Ghasemi et al., 2008; Tehrani et al., 2015), as well as in post-menopausal women with the metabolic syndrome (Chedraui et al., 2012) (Lines 376-379, References 71-74).

Q4. “Can diet intake regulate the levels of NO?”

A4. “Literature data demonstrated that beneficial health effects may be due to dietary nitrate intake from daily consumption of nitrate-rich foods. Vegetables and beetroot juice represent important sources of NO, since they contain an high amount of nitrate (Habermeyer et al., 2015), that is associated with the positive effects of this food groups against type-2 diabete and cardiovascular diseases (Ludberg et al., 2011; Carter et al.; 2010), and in blood pressure lowering and in vasoprotective effects (Kapil et al., 2010)” (Lines 380-385, References 75-78).

Reviewer 3.

In this paper, Budani and Tiboni highlights the role of Nitric Oxide (NO) in different aspects of ovarian biology. In particular, they review the NO regulations of steroidal hormones in the ovary, its modulation of the folliculogenesis and of the resumption of oocyte meiotic progression. Importantly, the authors evaluate also the aspects of NO activity in the regulation of embryo development in the context of in vitro fertilization, a subject which is of paramount importance in translational research. The topic is properly reviewed, and the introduction is fine and punctual. The very strength of this paper is that almost every section analyzing NO roles is supported by a table resuming the main findings and the paper from which they are taken, and two figures are also present to simplify the complexity of some molecular processes.

A1. Thanks to the reviewer for the positive comments.

Sincerely,

Professor Gian Mario Tiboni

Reviewer 2 Report

This manuscript provides a comprehensive review of the literature on the role of NO in the ovary. 

Overall, the manuscript reads well and is well organized. Authors have made significant efforts to summaries various studies on in vivo and in vitro models and the outcomes of the experiments. The discussion section towards the end of the manuscript is appreciable which provides the views of the author on the summarized literature. 

There are few comments that could be addressed, if within the scope of this manuscript:

  1. Is there any relationship shown for the role of NO, present in the ovary, with the ovarian cancer? 
  2. Could the inflammation regulated by NO have an affect on oocyte maturation or pregnancy?
  3. Any relationship to ovarian cyst? 
  4. beneficial and negative effects of NO in old vs young women OR pre-menopause vs post menopause?
  5. Can diet intake regulate the levels of NO?

Author Response

(The authors gave the same response as above.)

Reviewer 3 Report

In this paper, Budani and Tiboni highlights the role of Nitric Oxide (NO) in different aspects of ovarian biology. In particular, they review the NO regulations of steroidal hormones in the ovary, its modulation of the folliculogenesis and of the resumption of oocyte meiotic progression. Importantly, the authors evaluate also the aspects of NO activity in the regulation of embryo development in the context of in vitro fertilization, a subject which is of paramount importance in translational research.

The topic is properly reviewed, and the introduction is fine and punctual. The very strength of this paper is that almost every section analyzing NO roles is supported by a table resuming the main findings and the paper from which they are taken, and two figures are also present to simplify the complexity of some molecular processes.

Author Response

(The authors gave the same response as above.)
